# Modelling safe protocols for reopening schools during the COVID-19 pandemic in France

Laura Di Domenico[1], Giulia Pullano[1,2], Chiara E. Sabbatini[1], Pierre-Yves Boëlle[1] & Vittoria Colizza [1,3✉]

As countries in Europe implement strategies to control the COVID-19 pandemic, different options are chosen regarding schools. Through a stochastic age-structured transmission model calibrated to the observed epidemic in Île-de-France in the first wave, we explored scenarios of partial, progressive, or full school reopening. Given the uncertainty on children's role, we found that reopening schools after lockdown may increase COVID-19 cases, yet protocols exist to keep the epidemic controlled. Under a scenario with stable epidemic activity if schools were closed, reopening pre-schools and primary schools would lead to up to 76% [67, 84]% occupation of ICU beds if no other school level reopened, or if middle and high schools reopened later. Immediately reopening all school levels may overwhelm the ICU system. Priority should be given to pre- and primary schools allowing younger children to resume learning and development, whereas full attendance in middle and high schools is not recommended for stable or increasing epidemic activity. Large-scale test and trace is required to keep the epidemic under control. Ex-post assessment shows that progressive reopening of schools, limited attendance, and strong adoption of preventive measures contributed to a decreasing epidemic after lifting the first lockdown.

[1] INSERM, Sorbonne Université, Pierre Louis Institute of Epidemiology and Public Health, Paris, France. [2] Orange Labs, Sociology and Economics of Network and Services (SENSE), Chatillon, France. [3] Tokyo Tech World Research Hub Initiative, Institute of Innovative Research, Tokyo Institute of Technology, Tokyo, Japan. ✉email: vittoria.colizza@inserm.fr

Countries in Europe adopted different strategies to progressively phase out the strict restrictions put in place to curb COVID-19 pandemic[1–4]. They aimed to strike a delicate balance between reviving the economy and relieving social pressure while averting a potential resurgence of infections. Plans for lifting the first lockdown in Spring were quite heterogeneous in Europe[5], and a large debate sparked regarding the closure or reopening of schools. Italy and Spain chose to adopt restrictive and conservative solutions, keeping schools closed until September for precautionary reasons. Denmark and Norway reopened their primary schools. Austria allowed students to go back to school on May 18 with alternating classes, i.e., splitting students in two groups, each attending lessons during half of the week. Greece restarted classes with high schools first, followed by pre-schools and primary schools in June if epidemic conditions allowed.

On April 28, French authorities presented the exit strategies, with a progressive plan to reopen schools[6]. Pre-schools and primary schools were allowed to reopen on May 11, with classes limited to groups of 15 and based on voluntary attendance. Middle schools could follow one week later, but only in those departments weakly affected by the epidemic. Middle school students were asked to wear masks, differently from younger children. Reopening of high schools was to be decided in late May, depending on the epidemic evolution in each department. Universities would remain closed till September.

Assessing the risk that school reopening may have on the transmission of the epidemic faces a key challenge, as the role of children in COVID-19 spread is not yet well understood. Current evidence from household studies, contact tracing investigations, and modeling works suggest that children are less susceptible than adults, and more likely to become either asymptomatic or paucisymptomatic[7–11]. This may explain the very small percentage (<5%) of children in COVID-19 confirmed cases worldwide[12]. Their role in acting as source of infection remains unclear.

Epidemic data so far does not show the typical signature of widespread school outbreaks reported in past influenza pandemics, and responsible for driving transmission in the community[7–9]. However, such transmission could have gone unobserved because of (i) asymptomatic infections in children, (ii) testing restricted to symptomatic cases during the early phase of the outbreak, (iii) early school closure as reactive measure, or schools not in session because of holidays (e.g., in South Korea in January). A retrospective analysis of the Oise cluster in northern France showed evidence for large asymptomatic viral circulation in a high school, though initial case investigation and contact tracing had identified only two symptomatic cases (testing was not performed in absence of symptoms)[13].

Adolescents, however, may have a different role in driving the epidemic spread compared to younger children. Massive testing in Iceland and in the municipality of Vo', Italy, the initial epicenter of the Italian outbreak, showed that children under 10 years of age had a lower incidence of COVID-19 than adolescents and adults[14,15]. A second serological investigation performed retrospectively in the Oise cluster in the primary schools found similar results, together with evidence for lack of onward transmission after introductions of 3 infected children in primary schools[16]. In addition, contact tracing in South Korea showed that infection attack rates among household contacts of index cases were lowest when the index case was younger than 10 years old[17].

Multiple evidence therefore suggests that younger children may have a weaker role in COVID-19 transmission dynamics than adolescents. Accounting for this uncertainty, here we focused on the role of contacts at schools and the impact that different protocols for school reopening may have on the control of the epidemic in the successive months. We proposed scenarios of partial, progressive or full reopening of schools, with differential opening of pre-school and primary schools vs. middle and high schools, following the plan illustrated by the French Government[6] during the first lockdown. School reopening according to different protocols was compared with a scenario where all schools remained closed after lockdown ended, and moderate social distancing interventions[4], as well as extensive large-scale case tracing, testing, and isolation were in place. This scenario corresponds to a stable epidemic activity over time, after lifting the lockdown. The focus is on Île-de-France, the most affected region by the COVID-19 epidemic in France in the first wave.

This study was conducted in the lockdown phase, before its end in May, and was therefore based on a scenario analysis. Here, we also provide an ex-post assessment of the epidemic situation reported by data that became available after the initial submission.

## Results

**Age-structured transmission model and role of children**. We used a stochastic discrete age-structured epidemic model[18] based on demographic and age profile data[19] of the region of Île-de-France. Four age classes were considered: 0–11, 11–19, 19–65, and 65+ years old. The first class includes ages of students in pre-school and primary school, and the second class corresponds to students in middle and high school. We used social contact matrices measured in France in 2012[20] to account for the mixing, in the no interventions scenario, between individuals in these age groups, depending on the type of activity and place where contacts occur (household, school, workplace, transport, leisure, other) and the type of contact (physical or non-physical).

Transmission dynamics follows a compartmental scheme specific for COVID-19 (Supplementary Fig. 1), where individuals are divided into susceptible, exposed, infectious, hospitalized, in ICU, recovered, and deceased. The infectious phase is divided into two steps: a prodromic phase ($I_p$) and a phase where individuals may remain either asymptomatic ($I_a$) or develop symptoms. In the latter case, we distinguished between different degrees of severity of symptoms, ranging from paucisymptomatic ($I_{ps}$), to infectious individuals with mild ($I_{ms}$) or severe ($I_{ss}$) symptoms. Asymptomatic and paucisymptomatic individuals have a reduced transmissibility $r_\beta = 0.55$, as estimated in Ref. [21].

We considered the two classes of children to be half as susceptible as adults, and to become either asymptomatic or paucisymptomatic only, following refs. [7–9,21]. Viral load is similar across age classes[22] and across asymptomatic and symptomatic cases[15,23,24], however the risk of transmission was shown to vary with the presence and severity of symptoms[25]. Given the role of asymptomatic infection in high school students observed in the Oise cluster[13], we assumed that adolescents have the same reduction $r_\beta$ in transmissibility as adults in absence of symptoms[13–15]. We accounted for the weaker role of younger children in acting as source of infection by exploring four different values for their reduction in transmissibility: $r_\beta^{[0-11)} = 0.1, 0.25, 0.33, 0.55$. That is, their transmissibility is approximately 20%, 50%, 60%, 100% of the transmissibility of adolescents, respectively.

Intervention measures were modeled through modifications of the contact matrices, accounting for a reduction of the number of contacts engaged in specific settings[4,18]. The lockdown matrix was constructed assuming a certain fraction of workers not going to work (because of telework, closure of activity, caring for children not going to school, and other cases), school closure, 50% reduction of contacts established by seniors, and closure of non-essential activities (Table 1). We used data on mobile phone

**Table 1 Exit strategies following the lifting of the lockdown.**

| | School closure / reopening | Telework (=% of individuals not going to work) | Senior isolation | Closure non-essential activities | Case isolation | Adoption physical distancing (=% of individuals avoiding physical contacts) |
|---|---|---|---|---|---|---|
| Lockdown[4] | School closure | 70%[26] | Yes, with 50% contact reduction | Yes, 100% closure | No | 100% |
| Set of moderate interventions + case isolation[4] | School closure or Reopening through scenarios of Fig. 1 | 50% | Yes, with 30%[18] contact reduction | Yes, 50% closure | Yes, for 50% of cases (25% tested for sensitivity) | 0% (this parameter is explored in the ex-post analysis) |

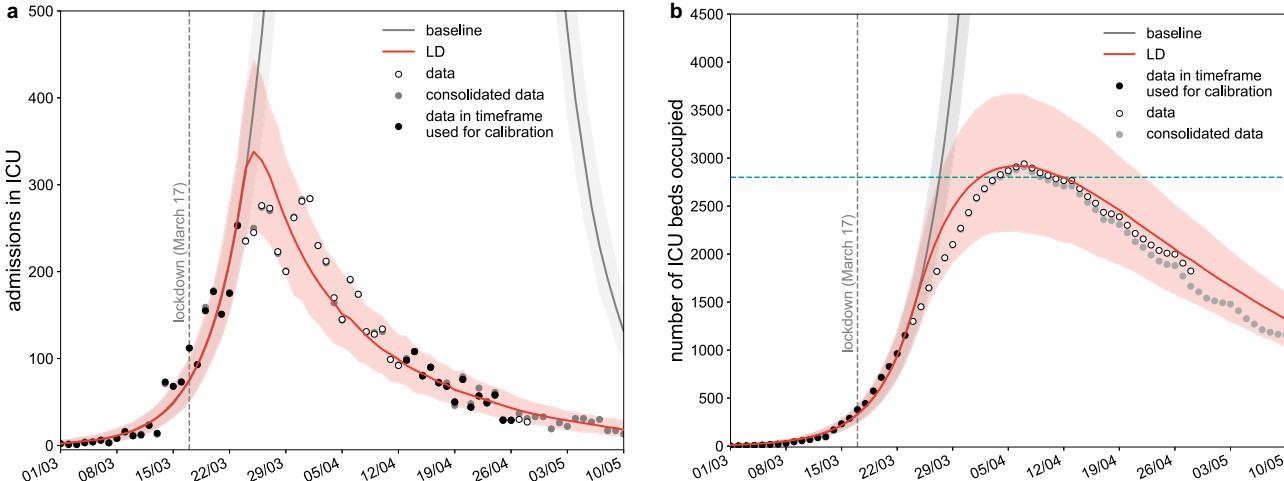

**Fig. 1 Simulated epidemic trajectories till May 11. a** Simulated daily incidence of admissions in ICU over time. **b** Simulated number of ICU beds occupied over time. The fitted curve slightly overestimates the data, likely because it does not account for the transfer of patients in intensive care to less affected regions. Vertical dashed line refers to the start of the lockdown; curves and shaded areas correspond to median and 95% probability ranges, obtained from $n=500$ independent stochastic runs; horizontal line refers to strengthened ICU capacity in the region to face the first COVID-19 wave; LD stands for lockdown. Black dots indicate data used for the calibration; gray dots indicate data that became available after the initial submission of this study.

trajectories in France to evaluate the change of mobility induced by lockdown[26] and informed the model with the estimated percentage of individuals in Île-de-France staying at home. Physical contacts engaged outside the household during lockdown were removed to account for the adoption of physical distancing.

The model was shown to capture the transmission dynamics of the epidemic in Île-de-France and was used to assess the impact of lockdown and exit strategies[4]. More details are available in the "Methods" section.

**Epidemic situation projected for May 11, 2020**. Calibrating the model in the lockdown phase to hospital and ICU admission data up to April 26, 2020, we estimated a drop of the reproduction number from $R_0 = 3.28$ [3.20, 3.39] (95% confidence interval) prior to lockdown[4] to $R_{LD} = 0.71$ [0.69, 0.74] during lockdown, in agreement with prior estimates[4,27,28]. Model projections indicate that by May 11 the region would experience 945 [802, 1076] new clinical cases per day (corresponding to 2391 [2025, 2722] new infections), 18 [11, 29] new admissions in ICUs, with an ICU system occupied at 47% [37, 57]% of strengthened capacity (Fig. 1). Our projections for ICU demand slightly overestimated the data, likely because they did not account for the transfer of patients in intensive care to less affected regions. These projections were obtained assuming that the reproduction number did not change throughout the lockdown phase. If the spreading

potential was 10% lower or higher than the estimated $R_{LD}$, e.g., corresponding to a decreased compliance to lockdown, the number of new clinical cases on May 11 was predicted to be 517 [423, 593] or 1648 [1432, 1853], respectively, corresponding to an ICU demand of 38% [30, 46]%, or 59% [48, 72]% of strengthened capacity.

**Scenarios for reopening of schools**. In ref. 4 we explored progressive exits from lockdown, with social distancing interventions of different degrees of intensity (strict, moderate, mild) coupled or not with case finding, testing and isolation. Following announcements by authorities, here we considered exit strategies as a combination of moderate interventions with efficient tracing, testing and isolation of cases (Table 1). Moderate interventions considered that 50% of adults would not go to work, 50% of non-essential activities remained closed, contacts of seniors were reduced by 30%[18], and contacts on transport were reduced according to presence of workers and reopening of activities. Scenarios assumed that physical contacts were fully restored phasing out lockdown (a different adhesion to physical distancing measure after lockdown was considered in the ex-post analysis). These social distancing interventions were combined with isolation of 50% of cases through a 90% reduction of their contacts, simulating the result of rapid and efficient tracing and testing of cases[4]. With schools closed, the resulting effective reproduction number corresponding to these interventions was $R_{MOD,SC}^{eff} = 1.02$

**Fig. 2 Protocols of school reopening.** The first set of scenarios considers the reopening of pre-schools and primary schools only, on May 11, through *Progressive (100%)*, *Progressive (50%)*, *Prompt (50%)*, and *Prompt (100%)* protocols. *Progressive (100%)*: progressive reopening up to 100% attendance, where 25% of students go back to school on the 1st week after lockdown is lifted, 50% on the 2nd, 75% on the 3rd, and 100% from the 4th week till summer holidays. *Progressive (50%)*: progressive reopening up to 50% attendance, where 25% of students go back to school on the 1st week after lockdown is lifted, and 50% from the 2nd week till summer holidays. *Prompt (50%)*: partial reopening with 50% attendance from May 11. *Prompt (100%)*: full reopening with 100% attendance from May 11. Colors indicate school levels (blue for pre-/primary schools, green for middle/high schools). Color gradient indicate student attendance (from lighter to darker, 25% to 100% at 25% incremental steps). The second set of scenarios considers the reopening of pre-schools and primary schools on May 11, only through *Progressive (100%)*, followed by the reopening of middle and high schools on June 8 through all 4 possible protocols. A sensitivity scenario assuming *Prompt (100%)* for pre-schools and primary schools is provided in the Supplementary Information. The third set of scenarios considers the reopening of all schools on May 11, with all schools following the same protocol.

[0.99, 1.06], leading to a stable epidemic activity over time. A sensitivity on the rate of case isolation under the moderate interventions scenario was also performed.

We simulated the reopening of schools on May 11 through three sets of four different scenarios (Fig. 2)—namely, progressive or prompt protocols at full or partial attendance, differentiated for type of schools (pre-school, primary, and middle, high school). The first set of scenarios considered the reopening of pre-schools and primary schools only, on May 11, whereas middle and high schools would remain closed till next school calendar. Progressive reopening was tested starting with an attendance of 25% in the first week that gradually increased over the following weeks, up to partial attendance (*Progressive 50%*) or full attendance (*Progressive 100%*). Prompt reopening was also considered, with 50% or 100% of students returning to school on May 11 (*Prompt 50%* or *Prompt 100%*). Protocols with 50% attendance envisioned for example a rotation of students every half of the week or every week, or considered 50% attending in the morning and 50% in the afternoon. The second set of scenarios considered the scenario *Progressive (100%)* for pre-school and primary schools starting May 11, coupled with the reopening of middle and high schools 4 weeks after (June 8) through progressive or prompt protocols at full or partial

attendance (i.e., as before, but for adolescents and starting on June 8). This set of scenarios was closer to the plan put in place by the French Government, accounting that Île-de-France was highly affected, so school opening for secondary school was expected to occur later compared to less affected regions. For sensitivity, we also considered the scenario *Prompt (100%)* for pre-school and primary school starting May 11, followed by the 4 scenarios for middle and high schools starting June 8 (Supplementary Fig. 5). Finally, the third set of scenarios assumed that all school levels, from pre-school to high school, would reopen after lifting the lockdown on May 11, through progressive or prompt protocols at full or partial attendance. Contacts at schools were proportional to school attendance in the various scenarios. All scenarios were compared to the situation where schools remain closed, under the moderate intervention scenario corresponding to an effective reproduction number $R_{\text{MOD,SC}}^{\text{eff}} = 1.02$ [0.99, 1.06].

**Impact on epidemic activity and health system.** If only pre-schools and primary schools reopened starting May 11 till the end of the school calendar, the projected number of new clinical cases at the start of summer holidays (July 5) was 2 to 2.4 times the number expected in the scenario with schools closed, depending

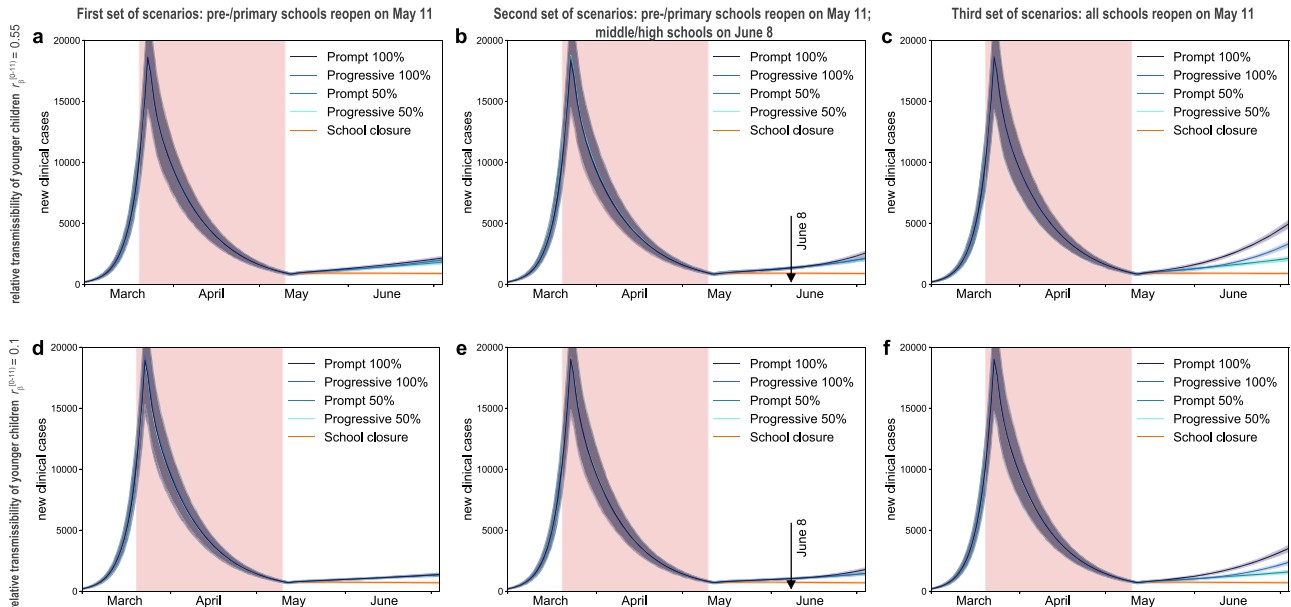

**Fig. 3 Simulated epidemic activity in scenarios with reopening of schools. a–c** Simulated daily number of new clinical cases assuming that only pre-schools and primary schools are reopened on May 11 through 4 different protocols (first set of scenarios, panel **a**), additionally considering the reopening of middle and high schools on June 8 (second set of scenarios, panel **b**), or assuming that all school levels reopen on May 11 (third set of scenarios, panel **c**). Four protocols (*Progressive (100%, 50%)*, *Prompt (100%, 50%)*) are compared to the school closure scenario. Curves and shaded areas correspond to median and 95% probability ranges, obtained from $n = 500$ independent stochastic runs. Results are obtained for a relative transmissibility of younger children $r_\beta^{[0-11]} = 0.55$, i.e., younger children are as infectious as adolescents. **d–f** As panels (**a–c**) assuming $r_\beta^{[0-11]} = 0.1$, i.e., transmissibility of younger children is about 1/5th of the one of adolescents. Results for other values of $r_\beta^{[0-11]}$ are reported in Supplementary Fig. 3. The red area indicates the lockdown phase. Results are obtained considering moderate social distancing interventions coupled with 50% case isolation.

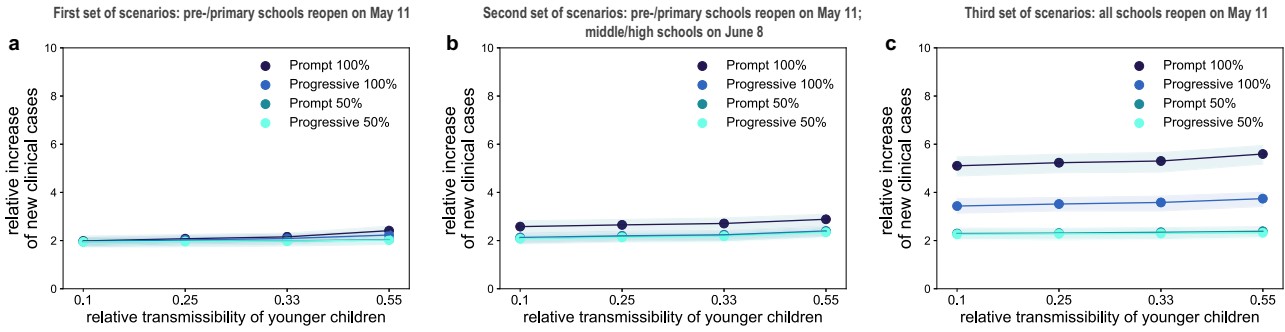

**Fig. 4 Impact of reopening schools on epidemic activity. a–c** Projected increase in the daily number of new cases relative to the school closure scenario on July 5 (start of summer holidays) as a function of the relative transmissibility of younger children, for different reopening protocols. Results are obtained considering moderate social distancing interventions coupled with 50% case isolation. Shaded areas correspond to 95% probability ranges around the median value, obtained from $n=500$ independent stochastic runs.

on the reopening protocol and the transmissibility of younger children (Fig. 3a,d and Fig. 4a). Though increasing, the epidemic would remain under control, with an expected maximum occupation of the ICU system on August 1 equal to 62% [54, 68]% of the foreseen 1500-bed capacity (Figs. 5a, 6a). No difference between protocols was observed when transmissibility of younger children was assumed to be lower than the transmissibility of adolescents ($r_\beta^{[0-11]} = 0.1, 0.25, 0.33$; Figs. 3d, 4a, and Supplementary Fig. 3). If $r_\beta^{[0-11]}$ is the same for both younger children and adolescents, maximum attendance mainly determines the increase of cases, whereas progressive and prompt protocols do not show substantial differences.

In addition, reopening middle and high schools starting June 8 would lead to an epidemic situation similar to the scenario with largest epidemic activity predicted for reopening lower

educational levels only (*Progressive (100%, 50%)* and *Prompt (50%)* of Figs. 3b, 4b compared to *Prompt (100%)* of Figs. 3a, 4a), if attendance was limited to 50% or if full attendance was reached through a progressive protocol. These scenarios would lead to a maximum ICU demand equal to 64% [56, 70]% of foreseen capacity. With full attendance in middle and high schools on June 8 (*Prompt (100%)* of Figs. 3b, 4b), the new number of clinical cases per day would be 2.6 to 2.9 times higher relatively to the school closure scenario at the start of the summer (median values; Fig. 4b), with an ICU occupation ranging from 54% [46, 60]% to 76% [67, 84]% by mid-summer (Fig. 6b), depending on younger children transmissibility. Results did not change if pre-schools and primary schools fully reopened on May 11 (Supplementary Figs. 5–7).

If all schools reopened on May 11 with full attendance, the increase in the number of new infections was predicted to lead to

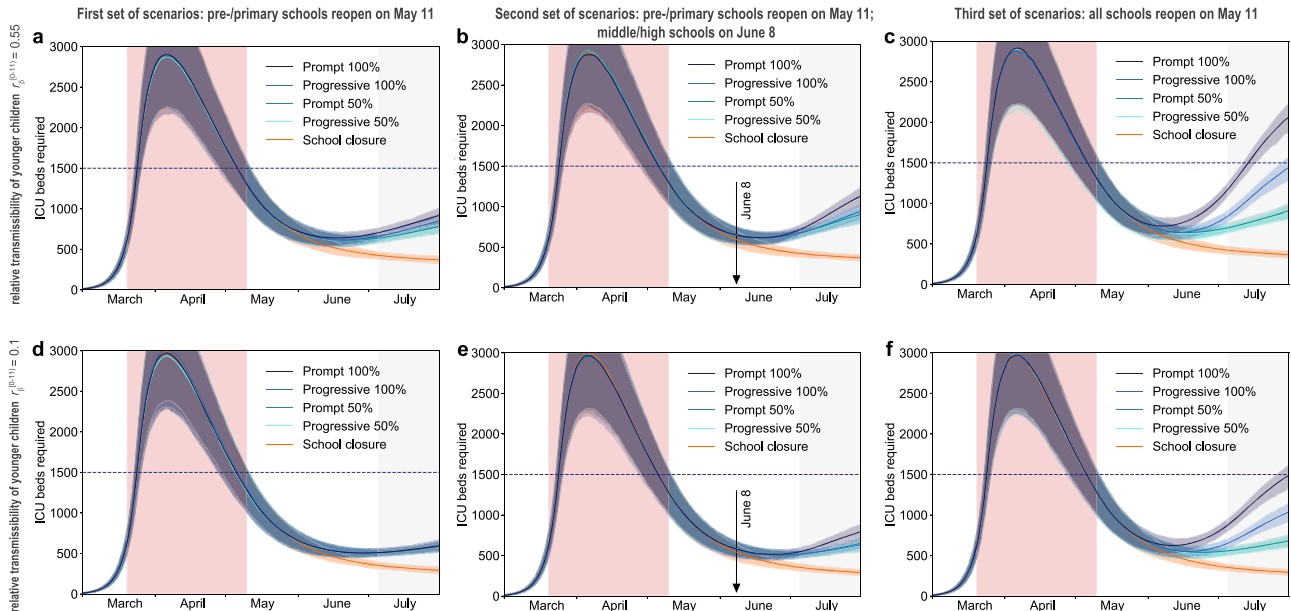

**Fig. 5 Simulated ICU occupancy in scenarios with reopening of schools. a–c** Simulated demand of ICU beds assuming that only pre-schools and primary schools are reopened on May 11 through 4 different protocols (first set of scenarios, panel **a**), additionally considering the reopening of middle and high schools on June 8 (second set of scenarios, panel b), or assuming that all school levels reopen on May 11 (third set of scenarios, panel **c**). Four protocols (*Progressive (100%, 50%), Prompt (100%, 50%)*) are compared to the school closure scenario. Curves and shaded areas correspond to median and 95% probability ranges, obtained from $n = 500$ independent stochastic runs. Results are obtained for a relative transmissibility of younger children $r_\beta^{[0-11]} = 0.55$, i.e., younger children are as infectious as adolescents. **d–f** As panels (**a–c**) assuming $r_\beta^{[0-11]} = 0.1$, i.e., transmissibility of younger children is about 1/5th of the one of adolescents. Results for other values of $r_\beta^{[0-11]}$ are reported in Supplementary Fig. 4. The red area indicates the lockdown phase; the gray area indicates summer holidays (month of July to show the delayed effect of the epidemic on ICU demand). Horizontal line refers to the foreseen 1500-bed ICU capacity in the region restored after the first wave emergency. Results are obtained considering moderate social distancing interventions coupled with 50% case isolation.

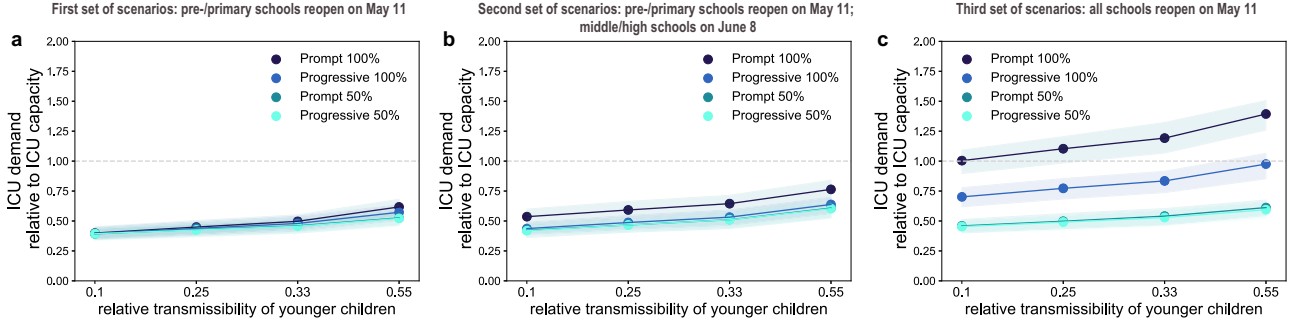

**Fig. 6 Impact of reopening schools on ICU occupancy. a–c** Projected ICU demand on August 1 relative to the foreseen 1500-bed ICU capacity in the region restored after the first wave emergency, as a function of the relative transmissibility of younger children, for different reopening protocols. Results are obtained considering moderate social distancing interventions coupled with 50% case isolation. Shaded areas correspond to 95% probability ranges around the median value, obtained from $n = 500$ independent stochastic runs.

saturation of the ICU capacity before the end of July (*Prompt (100%)* of Figs. 5c, 6c). Adopting a progressive reopening would still push the healthcare system to its limits by mid-summer, with ICU occupation reaching 98% [85, 107]% if younger children transmit as much as adolescents (*Progressive 100%*, Fig. 5c, Fig. 6c). To avoid this situation, limiting attendance to 50% per day would be key (maximum demand on ICU system as of August 1 expected to be between 45% [40, 51]% and 61% [55, 68]% of foreseen ICU capacity; Fig. 5c,f and Fig. 6c).

Among the scenarios avoiding oversaturation of the healthcare system, the model predicted a maximum of around 5000 cases per day in the region who would need to be promptly tested and put in isolation (*Progressive 100%*, third set of scenarios). Results were robust against a 10% increase of the reproduction number

during lockdown (Supplementary Fig. 8, with all scenarios avoiding full attendance of adolescents at school not exceeding ICU capacity). Reopening schools maintaining the epidemic under control would however require fast and aggressive tracing and testing of cases to allow their isolation. All scenarios obtained with 25% case isolation would overwhelm the ICU system by mid-summer (Supplementary Figs. 9–11). If contacts engaged by adolescents are reduced by 50%, the risk of reopening middle and high schools would be less pronounced (Supplementary Fig. 12) compared to the results of the main analysis.

**Ex-post assessment.** Starting May 11, schools in France reopened on a voluntary basis, and only a small percentage of students went back to school. The attendance registered in Île-de-France in June

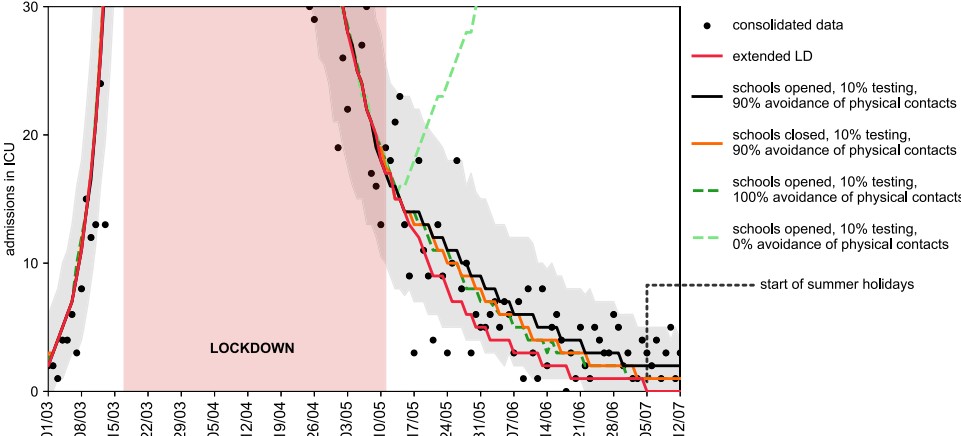

**Fig. 7 Retrospective analysis of the epidemic in the exit phase following lockdown.** Number of daily ICU admissions, comparison between data and different scenarios describing the post-lockdown phase. Red curve: lockdown maintained beyond May 11; black curve: scenario parameterized on data on interventions and attendance at school, with 90% avoidance of physical contacts; orange curve: as the black curve, with school closed; green curves: as the black curve, with full avoidance of physical contacts (dark green) or no respect of physical distancing (light green). The black curve corresponds to the maximum likelihood estimate of avoidance of physical contacts obtained by fitting ICU admissions data up to July 5, start of summer holidays. The red area indicates the lockdown phase. Curves indicate median values. Shaded areas around the curves indicate 95% probability ranges, obtained from $n = 500$ independent stochastic runs; they are shown only for the fitted scenario for the sake of visualization.

was 14.5% for pre-/primary schools[29], whereas middle and high schools remained closed at that time because of sustained viral circulation. Detection was estimated to be around 10% in the region for the overall period[18]. The epidemic continued to decrease in the two months of school reopening, after exiting lockdown, corresponding to a reproduction number of $R_{EXIT} = 0.83$ [0.81, 0.85]. The discrepancy with the scenarios analyzed before is mainly due to the conservative assumption of including all physical contacts after exiting lockdown, leading to an effective reproduction number around 1 with schools closed. i.e., leading to a stable epidemic activity over time instead of a decreasing one. Fitting our model to the observed trajectory of hospital admission exiting lockdown, we found that the maximum likelihood estimate for the percentage of the population avoiding physical contacts was 90% (Fig. 7). Such a large adhesion to recommendations on the use of preventive measures brought the reproduction number below 1. By simply altering adoption of physical contacts—while maintaining the protocols on schools, workplace, and non-essential activities unchanged—our results indicate that a large spectrum of dynamics are possible, from a continuing decrease of epidemic activity to an increased viral circulation leading to a rise in ICU admissions.

Scenarios presented before remain valuable to evaluate the impact of school opening under stable epidemic activity, for example at the start of the school calendar or after lifting again strict social distancing measures in the Fall/Winter 2020–2021[30].

## Discussion

Reopening schools after lifting lockdown is expected to lead to an increase in the number of COVID-19 cases compared to schools closed, even with lower transmissibility in children, yet protocols exist that would allow maintaining the epidemic under control without saturating the healthcare system. Starting from an epidemic scenario with an effective reproduction number around 1 if schools were closed, with pre-schools and primary schools in session starting May 11 ICU occupation would remain below the foreseen 1,500-bed capacity (at most 76% [67, 84]%), as long as middle and high schools limited students' attendance or reopened one month later. Healthcare system would exceed foreseen capacity (139% [126%, 151%]) if middle and high schools

reopened earlier in May accepting all students. Adopting a progressive protocol for adolescents starting May 11 would delay the increase in the epidemic activity, but largely engage the healthcare system by mid-summer (ICU occupancy at 98% [85%, 107%]). These findings are consistent across different assumptions on the relative transmissibility of younger children, however they require intensive large-scale tracing, testing, and isolation of cases, coupled with moderate social distancing interventions.

Easing exit strategies through progressive reopening of schools may help the preparation of schools to welcome younger children in the class. No substantial difference in the epidemic risk was predicted between progressive and prompt reopening of pre-schools and primary schools. In light of the 8 weeks left in the French school calendar after the first wave, full attendance of younger children in pre-schools and primary schools was thus predicted to be possible. This allowed resuming learning and development for all students in the age classes mostly in need[31]. Full attendance in middle and high schools was instead not recommended if the epidemic activity in the community was stable or increasing. Our findings are based on current evidence suggesting that higher school levels may become important settings for transmission[13], being then responsible for spreading the epidemic in the community. Virological evidence indicates that no significant difference in viral load is observed across age[22], or between symptomatic and asymptomatic infections[15,23,24]. Transmission, however, is mediated by symptoms and their severity[25], possibly explaining why younger children have a limited role as source of transmission[5,32]. Additional epidemiological and virological investigations are urgently needed to better characterize the role of children in the transmission dynamics of the disease, across age classes (e.g., also distinguishing between middle school and high school students), both at schools and in the community.

We considered moderate social distancing interventions[4], as they envision that a certain percentage of workers would resume their professional activity, including the partial reopening of commerce, while smart working would still be recommended. This scenario was in line with recommendations by the authorities[6] and was predicted to maintain the epidemic activity stable over time if extensive targeted tests were also implemented. In

absence of a widespread adoption of preventive measures by the population, our model predicted that about 5,000 infected individuals per day were to be isolated in Île-de-France at the largest epidemic activity (end of school calendar) in the scenarios allowing control of the epidemic. Targeting a maximum of 5% positivity rate of performed tests, as recommended by WHO[33], at least 100,000 tests per day would thus be required in the region at the peak of demand. This estimate exceeds the ballpark of expected regional capacity, considering the objective of 700,000 tests by week at national level originally announced by the Government and assuming a population-based distribution by region (i.e., about 18,000 tests per day in Île-de-France). Needs for testing, however, are greatly reduced in presence of large adhesion to recommendations for preventive measures that are able to mitigate the epidemic and lower peak demand[18]. Analyzing the epidemic in the post-lockdown phase once data became available, our results show that despite a low percentage of detection and isolation of infectious individuals[18], the epidemic continued to decrease, a dynamic that was also reported in other European countries[34]. Our results suggest that the continued decrease was due to a large adoption of preventive measures by the population, and to a minimal increase of contact activity in the community due to limited school attendance (and, specifically, only in the younger school levels). Our maximum likelihood estimate of 90% avoidance of physical contacts by the population is in line with estimates later provided by public health authorities through large-scale surveys for that period[18,35]. Communication on the sustained and continuative use of preventive measures is critical to manage the pandemic in the next months. At the same time, substantially more aggressive and targeted testing is needed as a strategy to control the epidemic[18]. Under the same social distancing strategy considered to lift lockdown, our findings confirm that a less efficient tracing, testing and isolation of cases was predicted to be unable to avoid a second wave[18].

Based on scenario analyses and retrospective assessment, our study provides a range of possible epidemic contexts where the opening of schools can be evaluated. Ultimately, the decision to reopen schools must depend on the trajectory of the epidemic, whether it is stable, increasing or decreasing as indicated by the reproduction number, and on the level of circulation of the virus as indicated by incidence in the community. In European countries, this whole range of situations was encountered from a decreasing epidemic at the beginning of the summer, to stability at the start of the school year in September and a rapid increase in cases in the fall[36]. The debate on school opening has become critical once again in this last period as strict restrictive measures were discussed and adopted to curb a second wave. Our analysis showed that reopening schools was possible under lockdown restrictions, as the effect on the epidemic would be only mildly affected[37]. France and Ireland adopted these approaches. However, we also found that specific attention should be taken for middle and high schools to reduce transmission, including reduced attendance, sanitary and cohorting protocols, and phased out breaks to avoid increasing circulation in the corresponding age classes. The use of regular screening could improve monitoring as recently proposed in Île-de-France[38].

The impact of school reopening on ICU occupancy is only visible after a certain delay, due to the natural progression of the disease, the spread among individuals in the community mainly leading to asymptomatic or subclinical forms of infection, and the long time period during which a patient requires intensive care. COVID-19 activity indicators estimated by the sentinel[39] and virological[18] surveillance systems need to be closely monitored to anticipate surge of patients. In addition, massive testing targeting cases and their contacts to break the chains of transmission through isolation will, at the same time, provide a reliable indicator to react in a more rapid and agile way to the evolving epidemic situation and revise social distancing recommendations.

ICU capacity was largely strengthened as an emergency response (approximately 2800 beds in Île-de-France compared to ~1200 in pre-pandemic conditions) to cope with the large influx of COVID-19 patients requiring critical care during the first wave. It required not only additional material (beds, respirators, rooms, etc.) but also personnel who was transferred from other medical specialties to support the increase in response. Exiting the first wave emergency, we envisioned that ICU capacity would be restored to lower levels for the following months, as the system was stretched to limits that are not sustainable in the long term. We considered a capacity of 1500 beds, i.e., a 25% increase compared to pre-COVID-19 epidemic size to re-establish almost routine conditions while accounting for the need to continue facing a pandemic situation for the next several months. If another emergency would occur, the system would need to be strengthened again to higher limits.

Our results were based on projections estimated on data up to April 28, 2020 and assumed that the reproduction number did not change throughout the lockdown phase. A 10% increase in the reproduction number than estimated from data, to account for uncertainty in the estimation or for reduced compliance to lockdown restrictions in the last weeks before May 11, did not affect our main results.

We did not consider explicitly the use of masks, as it was not generalized yet in the period under study and it was only required in public transports[6]. However, the effect of masks in reducing transmission in the community is implicitly accounted for in the calibration of the model in the ex-post analysis. In addition, avoidance of physical contacts is explicitly considered in the analysis once data became available. We modeled epidemic trajectories through summer to estimate the delayed impact that 8 weeks of schools in session in May and June may have on the hospital system in the month of July. However, we cannot accurately parameterize the model during summer, because of lack of contact data (we used Spring holiday contact data as a proxy instead) and of information about control measures and protocols for holidays that were still unknown at the time this work was conducted (we considered moderate social distancing measures still in effect, with no holidays). Also, we did not consider seasonal behavior in viral transmission[40], as this was still under investigation. Empirical contacts used in this study were measured in 2012, and no data are available to confirm that patterns have not been altered in more recent years. A reduction of 50% of contacts in adolescents, suggested by a recent study in the UK[41], would mitigate the increase in the epidemic activity when middle and high schools are reopened, as adolescents would play a less important role in the transmission dynamics relatively to other age classes.

We did not study reactive school closure as a means to slow down propagation[42,43], as our study was focused on the conditions allowing reopening for the last two months of the school calendar. We focused on Île-de-France region, following our previous work[4], and did not consider spatially targeted reopening within the region[44]. This work did not consider the circulation of newly emergent variants.

Given the heterogeneous situation in Europe regarding control strategies, and more specifically the opening or closure of schools[5], data gathered after the first wave becomes critical in the current and upcoming months as countries struggle to manage the pandemic during the winter season. Our findings, presenting different epidemic contexts corresponding to constant or decreasing viral circulation in the community, may help tailor interventions and inform decisions on the opening or closure of schools.

## Methods

**Parametrization of the model**. Parameters, values, and sources used to define the compartmental model are listed in Supplementary Table 1. Results of the main paper refer to the probability of being asymptomatic $p_a$=40%;[15] a sensitivity analysis on this value was performed in a previous work[4]. Time spent in each compartment was assumed to follow an exponential distribution. We used data on patient trajectories recorded in Île-de-France hospitals after admission to estimate time spent in hospital or ICU up to date of discharge or death. We fitted mixture and competing risks models to time to event data, taking into account censoring due to patients being still in the hospital at the time of analysis.

**Calibration of the model**. The model was calibrated to hospital and ICU admission data before and during lockdown through a maximum likelihood approach. Before lockdown we fitted the transmission rate per contact and the starting date of the simulation. During lockdown, we fitted the transmission rate per contact considering data in the interval April 13-26, 2020, to avoid fluctuations observed after lockdown entered into effect. More details are reported in the Supplementary Information.

**Case isolation**. Our model assumes that a percentage of infections is promptly tested and isolated to avoid onward transmission. Isolation corresponds to a 90% reduction of contacts. To account for a delay in tracing, testing, and self-isolation, we considered that infected individuals in their prodromic stage maintain their contacts as in the no-intervention scenario. The resulting effect of test-trace-isolate is therefore obtained by altering the number of contacts in the matrix associated to the detected infections, with a certain delay from the start of infectiousness.

**Holidays**. Summer vacations lasted from July 5 to August 31, 2020. The model cannot be easily parameterized during summer holidays because of lack of data and lack of information on recommendations for those months. For the simulations with reopening of schools, we used the number of contacts established by children during spring holidays, estimated by the social contact survey in France[20]. Holidays for adults were not modeled. We did not evaluate the epidemic trajectory during summer holidays (except for ICU occupancy due to the intrinsic delay in admission and occupation, see below), because of the above described uncertainties in the parameterization of the model.

**Evaluation**. Each scenario of school reopening was evaluated in terms of: number of clinical cases (i.e., cases with mild or severe symptoms) at the start of summer holidays (July 5, 2020) compared to the school closure scenario; ICU beds demand on August 1, to account for cases generated before summer holidays and the average delay due to disease progression. Current capacity of ICU beds in the region was largely strengthened during the first wave of the epidemic (approximately from 1200 to 2800 beds). To evaluate the school reopening scenarios, we considered an ICU capacity restored at 1500 beds in the months following the emergency, i.e., considering a 25% increase compared to standard pre-COVID-19 size. For each scenario, we performed 500 stochastic runs; median curves are displayed together with the associated 95% probability ranges.

**Sensitivity analysis**. We investigated social distancing measures based on moderate interventions coupled with a smaller rate of testing and isolation of cases (25%). We additionally performed the analysis also assuming that the reproduction number during lockdown is 10% lower or higher than the one estimated on current data. We evaluated the reopening of middle and high school starting June 8 (second set of scenarios), considering a full attendance of younger children starting May 11. Finally, we considered contact matrices with a 50% reduction of contacts in the [11,19) age class, to account for possible changes in the pattern of contacts measured in 2012, as suggested by a recent survey conducted in the UK[41].

**Ex-post assessment analysis**. We retrospectively compared ICU admission data in the post-lockdown phase with different exit scenarios. We considered (i) lockdown extended beyond May 11, (ii) a scenario parametrized on the observed attendance at schools and the estimated detection rate in the region, for varying adoption of avoidance of physical contacts and (iii) school closure parametrized with the estimated detection rate and the estimated avoidance of physical contacts. The second scenario is based on the reported attendance of 14.5% in pre-schools and primary schools in June, when middle and high schools were still closed[29]. Detection was parameterized based on estimates from another study[18], indicating that about 10% of cases were detected by the test-trace-isolate system in the period from May 11 to June 28, 2020. Avoidance of physical contacts was explored as a free parameter and fitted to ICU admission data up to July 5, through a maximum likelihood approach. Curves corresponding to 100% avoidance of physical contacts in the population and 0% avoidance are shown for comparison.

**Reporting summary**. Further information on research design is available in the Nature Research Reporting Summary linked to this article.

## Data availability

Aggregated hospitalization data are made available with the code at https://github.com/EPIcx-lab/COVID-19/tree/master/Impact_school_reopening. Contact data[20] and demographics data[19] are available at the references cited.

## Code availability

Analyses were carried out in Python 3.7.1. Code for the transmission model is available at https://github.com/EPIcx-lab/COVID-19/tree/master/Impact_school_reopening.

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

## Acknowledgements

This study is partially funded by: ANR projects SPHINX (ANR-17-CE36-0008-05) and DATAREDUX (ANR-19-CE46-0008-03); EU H2020 grants RECOVER (H2020-101003589) and MOOD (H2020-874850); REACTing COVID-19 modeling grant; INSERM-INRIA partnership on data science and public health. We thank Chiara Poletto, Alain Barrat, Juliette Paireau, and Santé publique France for useful discussions.

## Author contributions
V.C. conceived and designed the study. L.D.D., C.E.S., P.-Y.B. collected and analyzed the data. L.D.D., G.P. performed the simulations and numerical analysis. L.D.D., G.P., C.E.S., P.-Y.B., V.C. interpreted the results. V.C. drafted the Article. All authors contributed to the writing of the final version of the Article.

## Competing interests
The authors declare no competing interests.
