## [Peer Review File · Nature Communications]

REVIEWER COMMENTS

Reviewer #1 (Remarks to the Author):

An extensive study of different school opening scenarios in Northern France, with a specific focus on differentiating students less than 11 years old (pre- and primary school students) and students between 11-19 years olds (middle and high school students). Building on the evidence that the students less than 11 years old are less likely to be the source of transmission of the pandemic, a stochastic and age-based compartmental model is used to evaluate different opening scenarios. The main finding of the paper is that students less than 11 years old could start school earlier than older students, as according to the model presented, the healthcare system is able to handle the increased number of cases. The baseline scenario presented in the paper is to reopen all schools in May, and according to their model, the epidemic gets to a point to overwhelm the healthcare system. Besides these, the paper also makes general claims about the benefits of testing and large-scale trace, though I did find it quite hard (if at all) to tie the results to these claims about trace and testing.

I think the findings are (somewhat) novel, though the claims about trace and testing are not the focus of the results presented in the paper.

Major issues:

I find the main conclusion of the paper rather circular. I think the main premise of the paper is built on the assumption that younger students are less likely to transmit the disease. In a rather circular way, the main conclusion of the paper is that assuming the younger students (less than 11 years old) are less likely to transmit, reopening schools for them earlier than other older students is a good idea.

To (partially) address this issue, the authors perform sensitivity analysis, though they only assume transmissibility of at most 50% of other older age groups in the worst case.

Finally, I am unclear how the testing based isolation is incorporated in the compartmental model. I am not seeing an explicit compartment for the individuals who are tested positive and are self-isolating.

I think to make this paper more convincing more evidence needed to support the underlying assumption that young age groups are less likely to transmit the disease. I feel like the sensitive analysis is falling short to do so.

Minor Issues:

Line 51-53, why reduced number of cases due to the winter holidays is a confirmation only for the role of high-school? Were the other schools in session?

Line 72: Do age categories chosen in this paper fully align with the age categories in the referenced survey paper to obtain contact rates?

Line 74: What is the evidence that the contact rates are unchanged since 2012? What are the risks of using contact rate data from 2012?

Line 97: Could not find explicitly how mobility data is used to update the contact rates

Reviewer #2 (Remarks to the Author):

In this manuscript, the authors aim to evaluate the impact of school reopening as part of the exit strategies following the “lock-down” of France that has suppressed the first wave of COVID-19 epidemic in France. The authors utilized a highly detailed, age structured transmission model to characterize the transmission dynamic of COVID-19 in Ile-de-France, one of the hardest high regions in France. The model is informed by real-world survey data of contact patterns, mobility, intervention strategies, as well as relatively up-to-date information on the clinical and epidemiological features of SARS-CoV-2. The model is calibrated against observed hospitalization data and capable of reproducing the observed ICU admissions and occupations. The topic is timely relevant considering the current stages of the COVID-19 epidemics across the globe, especially for many countries who have successfully suppress the initial wave and are currently carrying out re-opening plans. The manuscript is well-written, and the conclusions are supported with thoroughly conducted simulation analysis. I would like to recommend the manuscript to be published as soon as the following comments have been addressed:

Major comments:

- The authors assume that the children under age 11 are equally susceptible to SARS-CoV2 as adults, but less infectious. However, as the science surrounding SARS-CoV-2 evolve quickly, we are accumulating evidence that children (especially primary school or younger) are less susceptible when compared to adults (see the meta-analysis of (1) for a nice summary on the evidence so far). However, very few studies are able to evaluate the relative infectiousness of children when compared to adults. I suggest the authors considering exploring the sensitivity of reduced susceptibility of children when compared to adults. In terms of relative infectiousness of children as already explored in the paper, it's fine to keep them in the paper but I do not have an issue if the authors decide to omit them in the manuscript for the sake of clarity and brevity, as a reduction in the infectiousness is already indirectly taken into account in the manuscript due to the high asymptomatic/paucisymptomatic rate of children.
- The authors consider a 50% of case isolation and contact reduction while exiting the “lockdown” (50% baseline, 25% in the sensitivity analysis). However, in the manuscript it's not clear at what stage of infection (Figure S1) the cases are being isolated. Mounting evidence suggest that SARS-CoV-2 is able to transmit during the pre-symptomatic phase. Thus, even if cases were isolated as soon as patients start developing symptom, transmission may already have occurred. The authors should make it clear how case isolation is implemented in the model.
- Would the authors be able to obtain the most up-to-date hospitalization and ICU data? If so, it would be of great interest to see which scenarios are most similar to the observation. If none of the scenario projections match the observation, it would be important to examine what interventions/components of the model are missing to reproduce the transmission dynamics.

1. R. M. Viner, O. T. Mytton, C. Bonell, G. J. Melendez-Torres, J. L. Ward, L. Hudson, C. Waddington, J. Thomas, S. Russell, F. van der Klis, J. Panovska-Griffiths, N. G. Davies, R. Booy, R. Eggo, medRxiv, in press, doi:10.1101/2020.05.20.20108126.

Minor points:

- “Reproductive number” -> “reproduction number”
- Line 77: “Transmission dynamics follows” -> “Transmission dynamics follow”
- Line 116-117: “Here we consider as exit strategy a combination...” -> “Here we consider exit strategies as a combination...”
- Line 124: “for type of” -> “four types of”
- Line 229: “by authorities” -> “by the authorities”
- Figure 2: it’s helpful for authors to label % of attendances in all table, even if they are implied by the color code”
- Figure 3: When referring to the “red areas” It would be great if the authors could use the same shade of red as in Figure 3.

Reviewer #3 (Remarks to the Author):

This is a timely paper addressing the impact of reopening schools in France following a large pandemic wave in Ile-de-France. The analysis is rigorous and the conclusions are actionable. Immediate reopening of all schools would be expected to overwhelm hospital capacity in the region. The study suggests that preschool and middle school children could be allowed back to school first followed by older students later in the school year, without undermining the integrity of the healthcare system.

Major Comments

1. While the methods appear sound, the article does not provide sufficient detail for replicating the analysis. Rather than relying on outside sources (especially ref. 21), the methods should be more fully described in a supplement. Specifically, the following assumptions/values should be specified:

- Population breakdown between students, general population, etc.
- Percent of individuals staying at home (line 98)
- Contact patterns for the different time periods
- Method for incorporating mobility data
- Method for modeling "isolation of 50% of cases through a 90% reduction of their contacts, simulating the result of rapid and efficient tracing and testing of cases". (line 120)
- Method for modeling various school reopening scenarios, including division of population and modifications to contact matrices. For example does 25% attendance mean that only 25% of that student group has their "school" contact matrix turned on? Or is it that 100% of students have their school contacts reduced to 25%?

2. The model assumes equal susceptibility for children and adults, but varies their transmissibility. However, recent studies suggest that children may be less susceptible than adults, perhaps even half as susceptible (Davies et al. 2020). How would this differential susceptibility alter your results?

3. Given that the school year has ended, the article should describe the real-world outcomes of opening schools and the observed epidemiology compares with the model predictions for the scenarios that most closely resemble what actually occurred.

4. The model assumes a symptomatic proportion of $p_a=20\%$. This seems low given more recent studies suggesting proportions closer to 50% (Lavezzo et al. 2020; Gudbjartsson et al. 2020).

Minor comments

1. The model initializes school reopening following the first pandemic wave. At that point, what are the estimated age-specific proportions of the population that are still susceptible?

2. Line 85 states that children are either asymptomatic or paucisymptomatic, but not fully symptomatic. Given the uncertainty surrounding the role children play in community transmission, it would be good to conduct a sensitivity analysis assuming that some or all infected children are as infectious as adult symptomatic cases.

3. Figure 1: The caption should clarify that LD and IDF stand for lockdown and Ile-de-France. For 1b, mention that the fitted curve appears to overestimate the data because some patients were moved to other regions (per line 106).

4. The progressive 50% and prompt 50% scenarios seem very similar. The impact of one week at 25% seems inconsequential, based on figures 3 and 4. Perhaps remove one of these two from the

main text (include in supplement) to simplify. If not, the figure captions should clarify that the curves are overlapping.

Reviewer #1 (Remarks to the Author):

An extensive study of different school opening scenarios in Northern France, with a specific focus on differentiating students less than 11 years old (pre- and primary school students) and students between 11-19 years olds (middle and high school students). Building on the evidence that the students less than 11 years old are less likely to be the source of transmission of the pandemic, a stochastic and age-based compartmental model is used to evaluate different opening scenarios. The main finding of the paper is that students less than 11 years old could start school earlier than older students, as according to the model presented, the healthcare system is able to handle the increased number of cases. The baseline scenario presented in the paper is to reopen all schools in May, and according to their model, the epidemic gets to a point to overwhelm the healthcare system. Besides these, the paper also makes general claims about the benefits of testing and large-scale trace, though I did find it quite hard (if at all) to tie the results to these claims about trace and testing.

I think the findings are (somewhat) novel, though the claims about trace and testing are not the focus of the results presented in the paper.

Major issues:

I find the main conclusion of the paper rather circular. I think the main premise of the paper is built on the assumption that younger students are less likely to transmit the disease. In a rather circular way, the main conclusion of the paper is that assuming the younger students (less than 11 years old) are less likely to transmit, reopening schools for them earlier than other older students is a good idea.

To (partially) address this issue, the authors perform sensitivity analysis, though they only assume transmissibility of at most 50% of other older age groups in the worst case.

In our study we use mathematical modeling to narrow down uncertainty. Evidence at the time of the study, which was also later confirmed, supported the hypothesis that younger children (students of pre-schools and primary schools) transmit less than adolescents when infectious. However, no estimate exists yet that quantifies their relative transmissibility. Given the uncertainty of this important parameter, we use mathematical models to simulate the transmission dynamics by assuming that transmissibility of younger children is ~20%, 50%, 60%, 100% the transmissibility of adolescents. As such, the model explores the full spectrum of possible values, including the assumption that younger children can transmit as much as adolescents. Two outcomes are possible: (i) the vast uncertainty in the input parameter produces equally vast uncertainty in the model outputs, thus preventing a possible assessment; (ii) despite the uncertainty in the input parameter, and once interventions and behavioural changes are accounted for, the model outputs show limited variation. In our study we find that we are in the second situation, and that -even assuming the worst conditions of transmissibility for younger children (i.e., they transmit as much as adolescents)- there exist protocols for school reopening that would maintain the epidemic under control.

The statement by the reviewer on the fact that our study only provides results for lower transmissibility of younger children is not correct. We believe that the possible confusion is due to the reduced transmissibility considered for adolescents ($r_{\text{beta}}=0.55$). Epidemic data show that adolescents are for the most part asymptomatic when infected. Also, large-scale outbreak investigations show that transmission risk increases with the presence of symptoms. Previous modeling fitting outbreak data in China estimated a reduced transmissibility for undocumented infections, i.e. asymptomatic infections but also likely paucisymptomatic infections that went undetected (all refs are cited in the paper). In our study we used this estimate to account for the reduced transmissibility of asymptomatic and paucisymptomatic infections, including infections in adolescents. That is, adolescents transmit as asymptomatic adults. For younger children, we explore values of transmissibility lower or equal to the transmissibility of adolescents. In other words, there are two aspects at play: a reduced transmissibility for asymptomatic supported by empirical data that applies to younger children and adolescents, and a possible further reduction in the transmissibility of younger children compared to adolescents, suggested by observations (though a quantitative estimation is still missing).

We have now made this explicit in the text.

Finally, I am unclear how the testing based isolation is incorporated in the compartmental model. I am not seeing an explicit compartment for the individuals who are tested positive and are self-isolating.

We did not explicitly model testing and self-isolation through a separate compartment, but we modeled its outcome by considering that a percentage of infectious individuals reduced their number of contacts by 90% to self-isolate as the result of testing positive. This is done by operating directly on the contact matrix for the % of tested positive and isolated infectious individuals. To account for a delay in testing and self isolation, we consider that individuals in their prodromic stage maintained their contacts as in the no intervention scenario. This approach was already used in our previous work where we assessed the impact of lockdown in Ile-de-France and proposed exit strategies (Di Domenico et al. BMC Medicine 2020). We have made this clearer in the main text.

I think to make this paper more convincing more evidence needed to support the underlying assumption that young age groups are less likely to transmit the disease. I feel like the sensitive analysis is falling short to do so.

The COVID-19 pandemic shows age-specific differences in the observed cases. This can be explained by the different role that children may have in the epidemic, including their number of contacts, their ability to transmit the disease, their likelihood of getting infected. Current knowledge from outbreak investigations and modeling works show that children have a lower propensity to show clinical symptoms, becoming either asymptomatic or paucisymptomatic once infected¹⁻⁴. Contact tracing investigations and modeling works support the evidence that individuals in the young age class are less susceptible to the infection^{1,5}. This was not considered in

the original version of our ms, as contrasting evidence still existed at that time; we are now presenting a revised version of our model that accounts for it.

Concerning the transmission, evidence collected in the last month suggests that there may be also a different behavior in COVID-19 transmission between younger children and teenagers, with the latter more likely to act as asymptomatic adults, whereas younger children are probably less infectious. This evidence emerges from cohort studies conducted in primary schools and in a high school in France^{6,7}. Serological investigations in one of the first large outbreaks in France detected in a high school showed a similar infection attack rate (IAR) between high school students and school personnel, higher than IAR in students' parents and siblings. This suggests that the high school was central in the outbreak dynamics. When scientists went back to perform serological investigations in the primary schools, they found lower IAR in students and staff, and evidence for lack of further transmission when the virus was introduced by 3 infected children in the primary schools. The lack of viral spread in primary schools, compared to the nearby high school, suggests that younger children may be less contagious. This is also supported by contact tracing data in South Korea, suggesting that IAR among household contacts of index cases were lowest when the index case was younger than 10 years old⁸.

Based on the evidence available at the time of the study, and also later confirmed, we considered a lower transmissibility of children younger than 10 years old as our starting hypothesis. Given the uncertainty that still exists on this aspect, we tested several hypotheses assuming as a worst case scenario that they would transmit like adolescents.

Minor Issues:

Line 51-53, why reduced number of cases due to the winter holidays is a confirmation only for the role of high-school? Were the other schools in session?

The winter holidays involved all school facilities and all school levels. At that time, only individuals with severe symptoms were tested and symptomatic individuals during contact tracing investigations. At the time of writing, a first serological outbreak investigation was available that was conducted in the high school linked to the outbreak. Authors found an important role of the high school in the outbreak dynamics due to the higher IAR in students and school staff compared to parents and siblings. Moreover, by reconstructing the curve of symptomatic cases based on declared onset of symptoms, they found that school holidays had an effect in reducing the number of symptomatic cases over time. For that outbreak, only transmission at the high school was known at that time (and at the time of writing the original version of this ms). Successive to the first serological analysis performed in the high school⁶, a second analysis was later performed in the nearby primary schools⁷. The authors found that transmission did not appear to have been impacted by the closure of schools for holidays on February 14. We have now rephrased the paragraph to account for this new evidence.

Line 72: Do age categories chosen in this paper fully align with the age categories in the referenced survey paper to obtain contact rates?

Yes. The available data⁹ provides contact rates in steps of 1-year age brackets; it is therefore possible to aggregate age groups in the most convenient way. We select the age classes [0-11), [11-19), [19-65), and 65+ years old in such a way as to be representative of the students in pre-school and primary school, students in middle and high school, adults and the elderly.

Line 74: What is the evidence that the contact rates are unchanged since 2012? What are the risks of using contact rate data from 2012?

A recent survey conducted in the UK within the BBC pandemic project¹⁰ suggested a decrease of nearly 50% in the average number of contacts made by teenagers (13–18 years) compared with the POLYMOD data, a large-scale endeavour for collecting social contact data in 8 European countries conducted between 2005 and 2006¹¹. It is still difficult to interpret these differences. They may genuinely indicate a change in social contacts of teens in the last decade, or they may also be the result of different data collection procedures. For example, in the most recent survey, individuals below 13 years old were not included in the study. To fully answer this question, the same experiment as POLYMOD should be repeated to have a robust assessment of possible changes. Clearly, social mixing conditions are now altered by the pandemic and by the recommendations on the use of preventive measures, such as physical distancing, avoiding physical contacts, avoiding crowded places etc. It will be hard to answer this question in the short time.

In our study, we considered the changes occurring in the population demographic profile from the time of the survey to today to account for changes in social contacts, but mixing rates were taken from the survey. This is a standard approach used in modeling studies (see e.g. Refs.^{12,13}). To account for the possible reduction in contacts in teens suggested by the BBC Pandemic study, we assessed the sensitivity of our results considering a baseline contact matrix with 50% reduction in the number of contacts engaged by individuals in the [11,19) age class. Fewer contacts established by adolescents lead to two effects: first, the calibration of the model on the observed epidemic in the pre-lockdown phase leads to a higher transmission rate per contact; second, adolescents play a less important role in the transmission dynamics relatively to the other age classes. Overall, the risk of the reopening of middle and high schools will be less pronounced compared to the results shown in the main analysis (see also below). These results are now included in detail in the supplementary information (Figure S12) and discussed in the main text.

Figure 1. Simulated impact on ICU occupancy of reopening schools with 50% reduction of contacts in the [11,19) age class. (a-c) Projected ICU demand on August 1 relative to the foreseen 1,500-bed ICU capacity in the region restored after the first wave emergency, as a function of the relative transmissibility of younger children, for different reopening protocols. Results are obtained considering moderate social distancing interventions coupled with 50% case isolation.

Line 97: Could not find explicitly how mobility data is used to update the contact rates

During lockdown, the only displacements allowed outside home were for essential work. Therefore, to account for how many individuals stayed at home because of telework or because they were workers in the job sectors impacted by the pandemic (e.g. employees at restaurants, cinemas, shops, etc.) or for other reasons (e.g. caring for children), we used the variation of mobility in the region pre- and during lockdown. This was measured from mobile phone trajectory data thanks to a collaboration with Orange, the main telephone operator in the country. For Île-de-France we found a reduction in the number of displacements of 70%¹⁴. Contacts at work and on transports were therefore reduced according to this percentage. This is now explained in detail in the Supplementary Information.

Reviewer #2 (Remarks to the Author):

In this manuscript, the authors aim to evaluate the impact of school reopening as part of the exit strategies following the “lockdown” of France that has suppressed the first wave of COVID-19 epidemic in France. The authors utilized a highly detailed, age structured transmission model to characterize the transmission dynamic of COVID-19 in Ile-de-France, one of the hardest high regions in France. The model is informed by real-world survey data of contact patterns, mobility, intervention strategies, as well as relatively up-to-date information on the clinical and epidemiological features of SARS-CoV-2. The model is calibrated against observed hospitalization data and capable of reproducing the observed ICU admissions and occupations. The topic is timely relevant considering the current stages of the COVID-19 epidemics across the globe, especially for many countries who have successfully suppress the initial wave and are currently carrying out re-opening plans. The manuscript is well-written, and the conclusions are supported with thoroughly conducted simulation analysis. I would like to recommend the manuscript to be published as soon as the following comments have been addressed:

Major comments:

The authors assume that the children under age 11 are equally susceptible to SARS-CoV- 2 as adults, but less infectious. However, as the science surrounding SARS-CoV-2 evolve quickly, we are accumulating evidence that children (especially primary school or younger) are less susceptible when compared to adults (see the meta-analysis of (1) for a nice summary on the evidence so far). However, very few studies are able to evaluate the relative infectiousness of children when compared to adults. I suggest the authors considering exploring the sensitivity of reduced susceptibility of children when compared to adults. In terms of relative infectiousness of children as already explored in the paper, it's fine to keep them in the paper but I do not have an issue if the authors decide to omit them in the manuscript for the sake of clarity and brevity, as a reduction in the infectiousness is already indirectly taken into account in the manuscript due to the high asymptomatic/paucisymptomatic rate of children.

(1) R. M. Viner, O. T. Mytton, C. Bonell, G. J. Melendez-Torres, J. L. Ward, L. Hudson, C. Waddington, J. Thomas, S. Russell, F. van der Klis, J. Panovska-Griffiths, N. G. Davies, R. Booy, R. Eggo, *medRxiv*, in press, doi:10.1101/2020.05.20.20108126.

We thank the reviewer for pointing out the building evidence around susceptibility in the younger age classes. This was not considered in the original version of the ms as it was still a matter of debate at that time. We have now reran simulations for all scenarios and updated all results considering that individuals below 19 years of age are half as susceptible as adults, following Ref. ¹. We kept however the variable transmissibility of younger children compared to adolescents, as additional evidence emerging during the review process confirmed that minors may transmit differently depending on their age^{6,7}. Serological investigations in one of the first large outbreaks in France detected in a high school¹ showed a similar infection attack rate (IAR) between high school

students and school personnel, higher than IAR in students' parents and siblings. This suggests that the high school was central in the outbreak dynamics. When scientists went back to perform serological investigations in the primary schools⁷, they found lower IAR in students and staff, and evidence for lack of further transmission when the virus was introduced by 3 infected children in the primary schools. The lack of viral spread in primary schools, compared to the nearby high school, suggests that younger children may be less contagious. This is also supported by contact tracing data in South Korea, suggesting that IAR among household contacts of index cases were lowest when the index case was younger than 10 years old⁸.

The authors consider a 50% of case isolation and contact reduction while exiting the "lockdown" (50% baseline, 25% in the sensitivity analysis). However, in the manuscript it's not clear at what stage of infection (Figure S1) the cases are being isolated. Mounting evidence suggest that SARS-CoV-2 is able to transmit during the pre-symptomatic phase. Thus, even if cases were isolated as soon as patients start developing symptom, transmission may already have occurred. The authors should make it clear how case isolation is implemented in the model.

To account for a delay in testing and self isolation, we consider that individuals in their prodromic stage maintain their contacts as in the no intervention scenario. This approach was already used in our previous work where we assessed the impact of lockdown in Ile-de-France and proposed exit strategies (Di Domenico et al. BMC Medicine 2020). We have made this clearer in the main text.

Would the authors be able to obtain the most up-to-date hospitalization and ICU data? If so, it would be of great interest to see which scenarios are most similar to the observation. If none of the scenario projections match the observation, it would be important to examine what interventions/components of the model are missing to reproduce the transmission dynamics.

We added a section on the retrospective analysis of the epidemic in the exit phase following lockdown. It is important to note, however, that our study was not meant to be a forecast of the upcoming evolution in time of the epidemic but rather a scenario analysis of what the impact of school reopening would be. For this reason, we originally chose a starting scenario (with schools closed) that kept the epidemic under control, i.e. corresponding to a continuous decrease of the hospitalizations. With the revisions accounting for a reduced susceptibility in the younger age classes, this intervention scenario now corresponds to $R \sim 1$, i.e. hospitalizations constant in time.

Starting May 11, schools in France reopened on a voluntary basis, and only a small percentage of students went back to school. The attendance registered in Île-de-France in June was 14.5% for pre-/primary schools¹⁵, whereas middle and high school remained closed at that time because of sustained viral circulation. Detection was estimated to be around 10% in the region for the overall period before school holidays¹⁶. The epidemic continued to decrease in the two months of school reopening, after exiting lockdown, corresponding to a reproduction number of 0.81 [0.79, 0.84]. The discrepancy with the scenarios analyzed before is mainly due to the conservative assumption of including all physical contacts after exiting lockdown. The maximum likelihood estimate in the

exit scenario corresponds indeed to 80% avoidance of physical contacts by the population, which is able to bring the reproduction number below 1. This estimate is in line with estimates provided by public health authorities through large-scale survey in that period¹⁷. By simply altering adoption of physical contacts – while maintaining recommendations on schools, workplace, non-essential activities – our results indicate that a large spectrum of dynamics are possible, from a continuing decrease of epidemic activity to an increased viral circulation leading to a rise in ICU admissions. Communication on the sustained and continuative use of preventive measures is critical to manage the pandemic in the next months.

Minor points:

We corrected these sentences following the reviewer's suggestion.

- “Reproductive number” -> “reproduction number”

Done.

- Line 77: “Transmission dynamics follows” -> “Transmission dynamics follow”

Done.

- Line 116-117: “Here we consider as exit strategy a combination...” -> “Here we consider exit strategies as a combination...”

Done.

- Line124: “for type of” -> “four types of”

Here we kept the original writing as the change would otherwise alter the meaning.

- Line 229: “by authorities” -> “by the authorities”

Done.

- Figure 2: it's helpful for authors to label % of attendances in all table, even if they are implied by the color code”

Done.

- Figure 3: When referring to the “red areas” It would be great if the authors could use the same shade of red as in Figure 3.

The red area in Fig3 and Fig 4 refers to the lockdown phase. To identify the summer break in the plots we used a different color (grey), to distinguish it from the lockdown phase.

Reviewer #3 (Remarks to the Author):

This is a timely paper addressing the impact of reopening schools in France following a large pandemic wave in Ile-de-France. The analysis is rigorous and the conclusions are actionable. Immediate reopening of all schools would be expected to overwhelm hospital capacity in the region. The study suggests that preschool and middle school children could be allowed back to school first followed by older students later in the school year, without undermining the integrity of the healthcare system.

Major Comments

1. While the methods appear sound, the article does not provide sufficient detail for replicating the analysis. Rather than relying on outside sources (especially ref. 21), the methods should be more fully described in a supplement. Specifically, the following assumptions/values should be specified:

We thank the reviewer for the comment. We explained in more details the methods in the main text and in the supplementary material.

- Population breakdown between students, general population, etc.

Four age classes are considered: [0-11), [11-19), [19-65), and 65+ years old. The first two age groups refer to students, [0-11) for pre-school and primary school, [11-19) for middle school and high school. School in France is compulsory up to the age of 16 and from 2020 also training between 16 and 18 years old is compulsory (it can be school, apprenticeship, etc)¹⁸, therefore we considered these age classes to correspond to the students. Then we consider adults, [19-65) years old and seniors, starting 65 years of age.

- Percent of individuals staying at home (line 98)

We used mobility data to inform the % of individuals staying at home. During lockdown, the only displacements allowed outside home were for essential work. Therefore, to account for how many individuals stayed at home because of telework or because they were workers in the job sectors impacted by the pandemic (e.g. employees at restaurants, cinemas, shops, etc.) or other reasons (e.g. caring for children), we used the variation of mobility in the region pre- and during lockdown. This was measured from mobile phone trajectory data thanks to a collaboration with Orange, the main telephone operator in the country. For Île-de-France we found a reduction in the number of displacements of 70%¹⁴. Contacts at work and on transports were therefore reduced according to this percentage.

- Contact patterns for the different time periods

Contact patterns during lockdown were informed from our previous work (Di Domenico et al. BMC Medicine 2020). Contact patterns after lockdown considered a scenario with moderate

interventions of social distancing (i.e. a partial (50%) reopening of non-essential activities, protection of seniors through a reduction of 75% of their contacts) coupled with more or less aggressive test-trace-isolate strategy (25% or 50% isolation of infected individuals, through a reduction of 90% of their contacts), with schools closed. Compared to this scenario, several protocols for school reopening were considered, where the key parameter was attendance each week. Given a value of attendance, the number of contacts in the school matrix was modified to account for the attendance of students in each school level.

- Method for incorporating mobility data

This corresponds to the estimation of the % of individuals staying at home, see above.

- Method for modeling “isolation of 50% of cases through a 90% reduction of their contacts, simulating the result of rapid and efficient tracing and testing of cases”. (line 120)

We report here the same answer to Reviewer #1.

We did not explicitly model testing and self-isolation through a separate compartment, but we modeled its outcome by considering that a percentage of infectious individuals reduce their number of contacts by 90% to self-isolate as the result of testing positive. This is done by operating directly on the contact matrix for the % of tested positive and isolated infectious individuals. To account for a delay in testing and self isolation, we consider that individuals in their prodromic stage maintain their contacts as in the no intervention scenario. This approach was already used in our previous work where we assessed the impact of lockdown in Ile-de-France and proposed exit strategies (Di Domenico et al. BMC Medicine 2020).

- Method for modeling various school reopening scenarios, including division of population and modifications to contact matrices. For example does 25% attendance mean that only 25% of that student group has their “school” contact matrix turned on? Or is it that 100% of students have their school contacts reduced to 25%?

Our matrices define average contacts at population level. Therefore, attendance of 25% corresponds to an average reduction of 75% in the number of contacts established at school by students belonging to that school level. For example, if pre-schools and primary schools are open with 25% attendance, and middle and high schools are closed, the elements of the school matrix would be equal to 0.25×0.25 in the child-child contacts, and to 0.25×0 in the child-adolescent contacts, and similarly for the other elements.

All the above elements are now better explained in the Supplementary Information.

2. The model assumes equal susceptibility for children and adults, but varies their transmissibility. However, recent studies suggest that children may be less susceptible than adults, perhaps even half as susceptible (Davies et al. 2020). How would this differential susceptibility alter your results?

Following this remark and the same remark raised by Reviewer #2 on the building evidence around susceptibility in the younger age classes, we have reran simulations for all scenarios and updated all results considering that individuals below 19 years of age are half as susceptible as adults, following Ref.¹. These are now presented in the main paper as the main parameterization of the model. Results obtained with the updated parameterization are qualitatively consistent with the results of the original analysis.

3. Given that the school year has ended, the article should describe the real-world outcomes of opening schools and the observed epidemiology compares with the model predictions for the scenarios that most closely resemble what actually occurred.

We added a section on the retrospective analysis of the epidemic in the exit phase following lockdown. It is important to note, however, that our study was not meant to be a forecast of the upcoming evolution in time of the epidemic but rather a scenario analysis of what the impact of school reopening would be. For this reason, we originally chose a starting scenario (with schools closed) that kept the epidemic under control, i.e. corresponding to a continuous decrease of the hospitalizations. With the revisions accounting for a reduced susceptibility in the younger age classes, this intervention scenario now corresponds to $R \sim 1$, i.e. hospitalizations constant in time.

Starting May 11, schools in France reopened on a voluntary basis, and only a small percentage of students went back to school. The attendance registered in Île-de-France in June was 14.5% for pre-/primary schools¹⁵, whereas middle and high school remained closed at that time because of sustained viral circulation. Detection was estimated to be around 10% in the region for the overall period before school holidays¹⁶. The epidemic continued to decrease in the two months of school reopening, after exiting lockdown, corresponding to a reproduction number of 0.81 [0.79, 0.84]. The discrepancy with the scenarios analyzed before is mainly due to the conservative assumption of including all physical contacts after exiting lockdown. The maximum likelihood estimate in the exit scenario corresponds indeed to 80% avoidance of physical contacts by the population, which is able to bring the reproduction number below 1. This estimate is in line with estimates provided by public health authorities through large-scale survey in that period¹⁷. By simply altering adoption of physical contacts – while maintaining recommendations on schools, workplace, non-essential activities – our results indicate that a large spectrum of dynamics are possible, from a continuing decrease of epidemic activity to an increased viral circulation leading to a rise in ICU admissions. Communication on the sustained and continuative use of preventive measures is critical to manage the pandemic in the next months.

4. The model assumes a symptomatic proportion of $p_a=20\%$. This seems low given more recent studies suggesting proportions closer to 50% (Lavezzo et al. 2020; Gudbjartsson et al. 2020).

The proportion of asymptomatic is still unknown and estimates vary wildly. A recent and living systematic review¹⁹ estimates this proportion to be 20% (95% CI 17-25) with a prediction interval of 3-67% in 79 studies. Besides the inherent difficulty in estimating this value, there are biases due to lack of follow-ups, identification of mild symptoms, differences in case definitions etc. that may alter this percentage. Additional systematic studies are needed to reliably answer this question. For this study, we kept therefore the value $p_a=20\%$ as in the original version. In a previous work on exit strategies (Di Domenico et al. BMC Medicine 2020), we showed the impact that a higher value would have.

Minor comments

1. The model initializes school reopening following the first pandemic wave. At that point, what are the estimated age-specific proportions of the population that are still susceptible?

Population is still largely susceptible after lockdown ends. Estimated proportions of individuals already infected in each age class as of May 11 are as follows: 1.7% [1.3, 2.1]%, 1.9% [1.4, 2.3]%, 4.3% [3.2, 5.3]%, 1.2 [0.9, 1.5]%, for younger children, adolescents, adults, and seniors respectively. The overall infected proportion in the population is 3.2% [2.3, 3.9]%. These values are in line with estimates from recent serological investigations performed in Italy²⁰, which underwent a similar epidemic as France in the first wave. Serological results are not yet available for France.

2. Line 85 states that children are either asymptomatic or paucisymptomatic, but not fully symptomatic. Given the uncertainty surrounding the role children play in community transmission, it would be good to conduct a sensitivity analysis assuming that some or all infected children are as infectious as adult symptomatic cases.

We did not test this assumption as evidence did not support it. Current knowledge from outbreak investigations and modeling works show that children have a much lower propensity to show clinical symptoms, and therefore they become either asymptomatic or paucisymptomatic once infected¹⁻⁴. Furthermore, the risk of transmission is estimated to increase with the presence of symptoms and with their severity²¹. A modeling work calibrated on an outbreak in China inferred that asymptomatic individuals and mild cases may transmit with half the rate of infection of more severe cases²². While there are few records showing severe disease in children, they represent a rare occurrence and for this reason are not considered in this work.

3. Figure 1: The caption should clarify that LD and IDF stand for lockdown and Ile-de-France. For 1b, mention that the fitted curve appears to overestimate the data because some patients were moved to other regions (per line 106).

Done.

4. The progressive 50% and prompt 50% scenarios seem very similar. The impact of one week at 25% seems inconsequential, based on figures 3 and 4. Perhaps remove one of these two from the main text (include in supplement) to simplify. If not, the figure captions should clarify that the curves are overlapping.

We thank the reviewer for this comment and clarified this in the figure captions.

References

- 1 Davies NG, Klepac P, Liu Y, Prem K, Jit M, Eggo RM. Age-dependent effects in the transmission and control of COVID-19 epidemics. *Nat Med* 2020; : 1–7.
- 2 Zimmermann P, Curtis N. Coronavirus Infections in Children Including COVID-19: An Overview of the Epidemiology, Clinical Features, Diagnosis, Treatment and Prevention Options in Children. *Pediatr Infect Dis J* 2020; **39**: 355–68.
- 3 Cai J, Xu J, Lin D, *et al.* A Case Series of children with 2019 novel coronavirus infection: clinical and epidemiological features. *Clin Infect Dis* DOI:10.1093/cid/ciaa198.
- 4 Bi Q, Wu Y, Mei S, *et al.* Epidemiology and transmission of COVID-19 in 391 cases and 1286 of their close contacts in Shenzhen, China: a retrospective cohort study. *Lancet Infect Dis* 2020; published online April 27. DOI:10.1016/S1473-3099(20)30287-5.
- 5 Zhang J, Litvinova M, Liang Y, *et al.* Changes in contact patterns shape the dynamics of the COVID-19 outbreak in China. *Science* 2020; **368**: 1481–6.
- 6 Fontanet A, Tondeur L, Madec Y, *et al.* Cluster of COVID-19 in northern France: A retrospective closed cohort study. *medRxiv*. 2020; published online April 23. <https://www.medrxiv.org/content/10.1101/2020.04.18.20071134v1> (accessed April 29, 2020).
- 7 Fontanet A, Grant R, Tondeur L, *et al.* SARS-CoV-2 infection in primary schools in northern France: A retrospective cohort study in an area of high transmission. *medRxiv*. 2020; published online June 29. <https://www.medrxiv.org/content/10.1101/2020.06.25.20140178v2>.
- 8 Park YJ, Choe YJ, Park O, *et al.* Contact Tracing during Coronavirus Disease Outbreak, South Korea, 2020 - Volume 26, Number 10—October 2020 - *Emerging Infectious Diseases journal* - CDC. DOI:10.3201/eid2610.201315.
- 9 Béraud G, Kazmierczak S, Beutels P, *et al.* The French Connection: The First Large Population-Based Contact Survey in France Relevant for the Spread of Infectious Diseases. *PLOS ONE*. 2015; **10**: e0133203
- 10 Klepac P, Kucharski AJ, Conlan AJ, *et al.* Contacts in context: large-scale setting-specific social mixing matrices from the BBC Pandemic project. *medRxiv* 2020; : 2020.02.16.20023754.
- 11 Mossong J, Hens N, Jit M, *et al.* Social Contacts and Mixing Patterns Relevant to the Spread of Infectious Diseases. *PLoS Med* 2008; **5**: e74.
- 12 De Luca G, Kerckhove KV, Coletti P, *et al.* The impact of regular school closure on seasonal influenza epidemics: a data-driven spatial transmission model for Belgium. *BMC Infect Dis* 2018; **18**: 29.
- 13 Arregui S, Aleta A, Sanz J, Moreno Y. Projecting social contact matrices to different demographic structures. *PLOS Comput Biol* 2018; **14**: e1006638.
- 14 Pullano G, Valdano E, Scarpa N, Rubrichi S, Colizza V. Population mobility reductions during COVID-19 epidemic in France under lockdown. *medRxiv* 2020; published online June 1. <http://medrxiv.org/lookup/doi/10.1101/2020.05.29.20097097> (accessed June 5, 2020).
- 15 Déconfinement phase 2 : point de situation au 28 mai. *Ministère Educ. Natl. Jeun.*

<https://www.education.gouv.fr/deconfinement-phase-2-point-de-situation-au-28-mai-303813> (accessed July 17, 2020).

- 16 Pullano G, Di Domenico L, Sabbatini CE, *et al.* Underdetection of COVID-19 cases in France in the exit phase following lockdown. *medRxiv* 2020; published online Aug 12. <https://www.medrxiv.org/content/10.1101/2020.08.10.20171744v1>.
- 17 Covid-19 : une enquête pour suivre l'évolution des comportements et de la santé mentale pendant l'épidémie. /etudes-et-enquetes/covid-19-une-enquête-pour-suivre-l'évolution-des-comportements-et-de-la-sante-mentale-pendant-l-epidemie (accessed July 21, 2020).
- 18 Système éducatif en France. Wikipédia. 2020; published online Aug 7. https://fr.wikipedia.org/w/index.php?title=Syst%C3%A8me_%C3%A9ducatif_en_France&oldid=173636154 (accessed Aug 9, 2020).
- 19 Buitrago-Garcia DC, Egli-Gany D, Counotte MJ, *et al.* Asymptomatic SARS-CoV-2 infections: a living systematic review and meta-analysis. *medRxiv* 2020; : 2020.04.25.20079103.
- 20 Indagine sierologica su Covid-19 condotta da Ministero della Salute e Istat. 2020; published online Aug 3. <https://www.istat.it/it/archivio/242676> (accessed Sept 5, 2020).
- 21 Luo L, Liu D, Liao X, *et al.* Modes of contact and risk of transmission in COVID-19 among close contacts. *medRxiv*. <https://www.medrxiv.org/content/10.1101/2020.03.24.20042606v1>.
- 22 Li R, Pei S, Chen B, *et al.* Substantial undocumented infection facilitates the rapid dissemination of novel coronavirus (SARS-CoV2). *Science* 2020; published online March 16. DOI:10.1126/science.abb3221.

REVIEWERS' COMMENTS

Reviewer #2 (Remarks to the Author):

The authors have adequately addressed all my previous comments and I recommend the article to publish at Nature Communication.

Reviewer #3 (Remarks to the Author):

The authors addressed all of our major concerns either through revisions or clear rebuttals. This is a timely and rigorous contribution.